# Gender and cultural aspects of brucellosis transmission and management in Nakasongola cattle corridor in Uganda

Christine Tricia Kulabako[1,2]*, Stella Neema[1], Lesley Rose Ninsiima[2], Jörn Klein[3], Lydia Namakula Nabawanuka[4], James Muleme[2,4], Javier Sánchez Romano[5], Peter Atekyereza[1]

1 Department of Sociology and Anthropology, Makerere University, Kampala, Uganda, 2 Department of Biosecurity, Ecosystems and Veterinary Public Health, Makerere University, Kampala, Uganda, 3 University of South Eastern Norway, Porsgrunn, Norway, 4 Department of Disease Control and Environmental Health, Makerere University, Kampala, Uganda, 5 Department of Medical Biology, The Artic University of Norway, Tromsø, Norway

* kulabakokristin1@gmail.com

## Abstract

### Background

Brucellosis is a zoonotic disease with significant public health and economic effects on societies. In Uganda, brucellosis is endemic and a primary contributor in livestock productivity losses. This is more worrisome for populations in the cattle corridor with high reliance on cattle and milk for nutritional value and symbol in social relations and identity. The community's social construction may affect comprehension of brucellosis hence leading to exposure and increased vulnerability to transmission. Despite brucellosis' high prevalence in the cattle corridor, little attention has been paid to its social construction. Hence, this study explored the interplay between gender dynamics, vulnerability and social construction of brucellosis transmission, in consideration of the unique socio-cultural context that characterizes cattle corridor populations.

### Methods

Using an exploratory qualitative approach, the study was conducted in Nakasongola cattle corridor within three sub counties; Nabiswera, Nakitoma and Wabinyonyi using key informant interviews (KIIs) and focus group discussions (FGDs). Purposive sampling was used to identify participants for the four FGD [8–12] each from a subcounty though one was combined and 15 KIIs. Data were collected using face -to -face interviews with an interview guide that was structured using the Socio Ecological Model of Human Behaviour framework (SEMHB) constructs. Thematic analysis was conducted in NVivo 12 Pro incorporating both deductive (guided by the SEMHB) and inductive approaches (guided by the data).

**Data availability statement:** All relevant data are available within the paper and its supporting information files.

**Funding:** Funding was obtained from Norwegian Agency for Development Cooperation (NORAD) through the NORHED-II Program and the project Climate Change Infectious Diseases - A One Health Approach (CIDIMOH), grant number 68802. The funders had no role in study design, data collection and analysis, decision to publish, or preparation of the manuscript.

**Competing interests:** The authors declare no competing interests.

## Findings

The study identified important themes under each SEMHB influence level (Individual, Interpersonal, Community and Societal level). The study indicates that social composition and role distribution are driven by social and cultural expectations and significantly contribute to exposure and vulnerability to infection in the cattle corridor. For instance, it is paramount that women undergoing marriage preparations to be fed on raw milk for a certain period prior to their ceremony to enhance beauty. Also, important to note that use of personal protection to assist births is viewed by the community as opposing cultural norms, creating a perception of detachment from the highly valued cattle. Another noteworthy finding is the level of knowledge on brucellosis in terms of symptoms, transmission route, prevention and treatment at the interpersonal level. Furthermore, findings show practices such as the consumption of raw milk and assisted births, as being rooted in the social cultural norms, hence critical for transmission of brucellosis. At the community and organizational levels, the findings indicate an inadequate level of knowledge sharing and reluctance towards preventive measures as structural factors for the transmission of brucellosis and are ingrained in family and power relations.

## Conclusion

The findings highlight that the social construction of brucellosis transmission is rooted in gender roles, social- cultural and power structures highlighting the influence of living process and spaces, at the different societal levels. Such complex dynamics play a critical role in determining individuals' susceptibility to infection as well as transmission potential of the disease-causing agent in cattle keeping communities. The gendered induced vulnerabilities related to the socio-cultural norms and familial roles, also play an important role in the exposure and spill over at the individual, interpersonal and community levels. The insufficient knowledge-sharing and reluctance to adopt preventive measures emerge as structural contributors to the persistence of brucellosis and other emerging zoonoses. These factors, intertwined with family dynamics and power relations, call for targeted interventions that address both individual behaviors and broader socio-cultural and institutional barriers to effective disease management and prevention. Conversely, policies that align with the community's social construction, gender and context are more likely to be feasible, adopted and sustained by the affected population.

## Introduction

Brucellosis is a zoonotic disease that has adverse effects globally with a conservative global annual incidence of 1.6–2.1 million new cases [1]. Brucellosis transmission occurs through multiple pathways, including inhalation and dermal entry of bacteria through cut skin [2], as well as consumption of unpasteurized milk or undercooked meat and direct contact with infected animals; with occupational exposure

posing a higher risk. People who work closely with animals, such as veterinary healthcare professionals, farmers, pastoral communities, abattoir workers, and laboratory personnel, are at a high occupational risk of contracting brucellosis [3].

Brucellosis is considered endemic in areas with limited resources like Uganda and three (*Brucella melitensis*, *Brucella abortus* and *Brucella suis)* of the brucellosis species have been identified as virulent both to livestock and humans [4]. In Uganda, brucellosis has been listed as fourth among the seven priority zoonoses for control and ultimate eradication [5]. A number of serological surveys conducted in cattle in different districts in Uganda have reported varying prevalence rates [6,7]. Brucellosis is endemic in areas of south western Uganda and among agro-pastoralists with seropositivity 11% in humans and high prevalences of 14% in food animals like cattle and goats [7,8]. Even though the global zoonotic risk has long been acknowledged, high burden among the livestock farmers within the cattle corridor is high [9].

Livestock farming is essential to the livelihoods of the population in Uganda especially in the cattle corridor and other regions of Africa [10,11] with both economic and social-cultural status and significance. Farmers in these communities rely on their herds of mixed livestock species for their daily needs in terms of food, income [12], and social interactions, yet zoonotic illnesses like brucellosis have a disastrous impact on these communities. Although existing literature have emphasized the biological understanding of brucellosis providing insight into its causative agents [13], transmission modes [14], and clinical manifestations [15], there is a critical gap in understanding the societal constructs and behavioral factors that influence disease occurrence and transmission. The dynamics of transmission factors for diseases like brucellosis are deeply embedded within societal structures and constructions. Social constructs such as, gender dynamics, roles, cultural practices, beliefs, values and other societal determinants shape an individual's unique susceptibility to the disease; impact of the illness on their livelihood and the wellbeing of their household. In addition, such dynamics can also impact individuals' engagement with healthcare systems, disease exposure, and overall health outcomes. For instance the social construction of gender identity influences anticipated behaviors of individuals, shaping both societal perceptions of them and their self-perception [16]. Hence the gender-specific interactions across the livestock continuum, from on-farm activities to production, processing, and post-consumption, significantly influence exposure and disease transmission risk [17]. However, this gap between social constructs and biological understanding can hinder effective disease management thus highlighting the need to consider these interconnections in disease prevention and management strategies.

Acknowledging this gap, the study utilized a theoretical framework that has proven effective in the creation and enhancement of disease prevention initiatives; the Social Ecological Model of Human Behaviour(SEMHB) [18]. The SEMHB is a multi-level hierarchical organization of systems centered on the individual that was designed to explain the drivers of human behavior based on the hierarchical interactions of biological systems [19]. The model recognizes that individuals' health-related choices, while being their own responsibility, are influenced by various factors functioning at multiple levels [20]. This approach therefore takes into account the intricate interactions that exist between societal, interpersonal, communal, and individual elements [21]. When applied to brucellosis disease transmission and management, gender becomes a key element that influences each layer of this SEMHB model through recognizing gender roles, social expectations, relationships and power dynamics; and how they shape behaviour and access to resources. The model assists us in gaining a deeper comprehension of the aspects influencing behavior and what could be required to change it. Thus, rather than displacing other behavioral theories, this theory frames the research in a more comprehensive manner. This study, therefore, aimed to explore the interplay between gender dynamics, vulnerability and social construction in the management of brucellosis, in consideration of the unique socio-cultural context that characterizes cattle corridor populations in Nakasongola District.

## Methods

### Study area

The study was conducted in Nakasongola district, which is centrally located in Uganda, approximately 114km from Kampala city. This district spans an area of 3102.4 sq km, with a population density of 63.5 persons per sq km [22] and a total population of 217,648 persons (51.3% male, 48.7% female) [23]. Nakasongola, is part of the Uganda's cattle corridor, and

it's characterized by its semi-arid and dry sub-humid climatic conditions receiving low and unreliable rainfall (875–1000mm annual rainfall) and with temperatures ranging from 17 to 32°C [24]. The district is dominated by the Bantu speaking group known as Baruuli, though they co-exist with other ethnicities like the Karamojong, Luo, Bakenyi and Banyoro. The economy is primarily driven by pastoralism and livestock farming, with the majority of the population living in the rural area. Geographically, the district is located between longitudes 21°55'E and 32°50'E, and latitudes 0°55'N and 1°40'N. Nakasongola has 9 sub-counties, 45 parishes and 364 villages. However, the research took place in three sub-counties of Nabisweera, Wabinyonyi and Nakitoma Fig 1, which were selected because of the higher number of herds (12–20%) and prevalence of brucellosis (18%) compared to other sub-counties [23,25].

## Study design, study population and sampling strategies

This study employed an exploratory qualitative design and was conducted between December 22, 2023, and January 24, 2024, in three purposively selected sub-counties: Nabiswera, Wabinyonyi, and Nakitoma, located in Nakasongola District. These sub counties were chosen due to their high livestock population and significant prevalence of brucellosis [26]. The selection also aimed to provide a geographically representative overview of the diverse communities across various locations, enhancing the findings' transferability. The focus on pastoralist communities who are commonly known as the *balaalo* was intentional, and aligns with the literature indicating an increase of exposure to brucellosis within these communities [27]. The distinction was significant since experiences can be influenced by cultural attitudes held by people of different backgrounds.

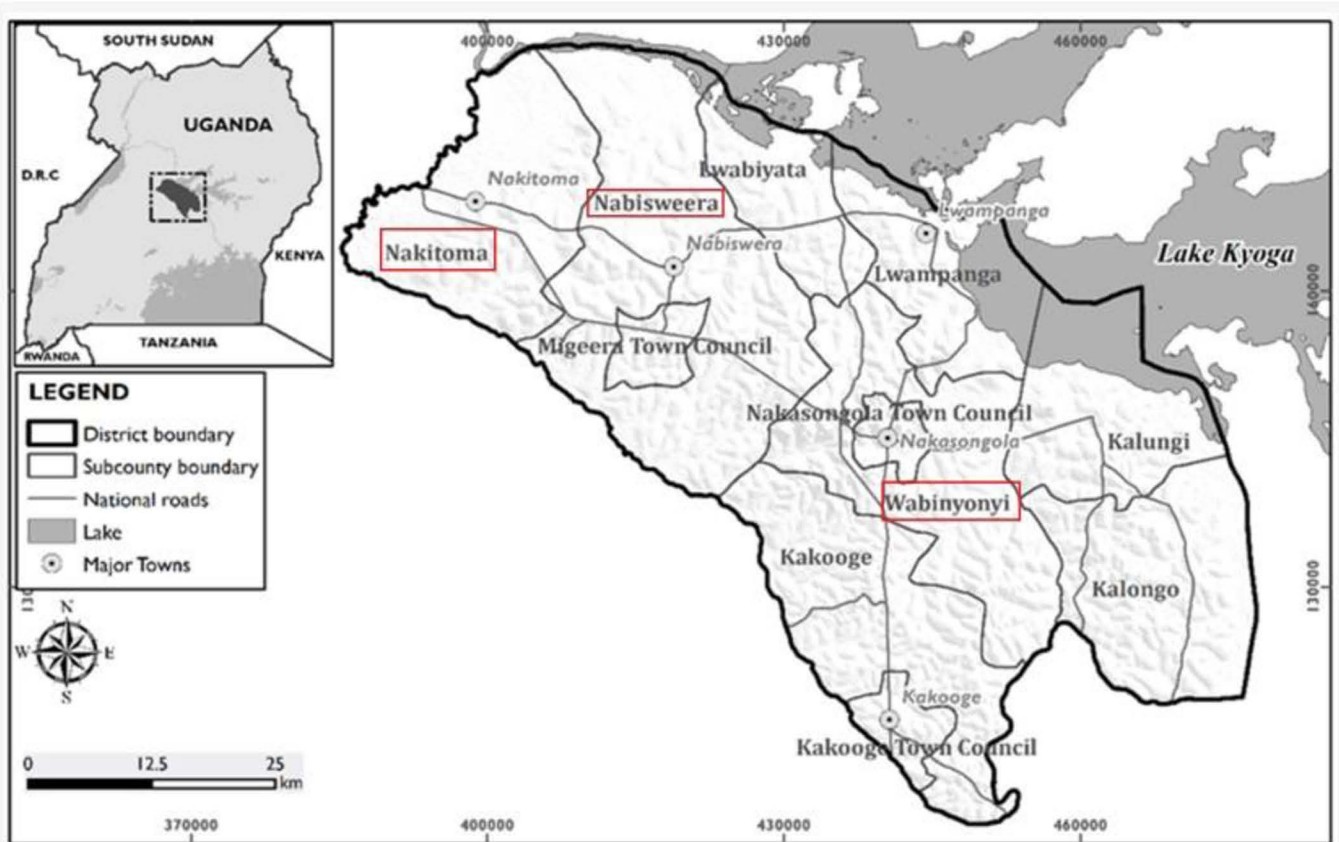

**Fig 1. A map of Nakasongola showing its administrative units.** Sub-counties were the interviews were conducted are highlighted in red.

Purposive sampling was used to select both the participants of the focus group discussions and key informants using maximum variation strategy and the sample size was determined by participant saturation [28], a recruitment matrix was created to keep track of the maximum variation sampling, with the aim of at least one participant in each matrix cell. Participants were approached physically and asked about their willingness to participate in the study thus recruitment was based on knowledge, experience, and interest in the study with the aim of gaining a wide range of perspectives. We had a thorough discussion about the purpose of the research, pointing out the data to be collected, the role of the researcher and the reasons for their recruitment. This approach enhanced rapport among the participants and the researcher. All those contacted agreed to participate and written informed consent was obtained from all the participants before commencement.

## Data collection and procedure

Data was collected using key informant interviews and focus group discussions (S1 File). A total of 4 Focus Group Discussions (FGDs) with a range of [8–12] participants and 15 Key Informant Interviews (KIIs) as shown in Table 1. The variation in the number of members within the focus group discussions (FGDs) enhances the depth of discussions, fosters meaningful interactions, and promotes the inclusion of diverse perspectives [29]. The KIIs were selected from all the participating sub-counties (Nabiswera, Nakitoma, and Wabinyonyi) with each providing five participants for representativeness and for offering thorough insights into the social factors for transmission of brucellosis. The KIIs participants included medical personnel, village health teams (VHTs), local veterinary officers social and cultural leaders to understand different perspectives of brucellosis management. As a pre-requisite, participants had to provide written informed consent and had to be knowledgeable and experienced with brucellosis to qualify for participation. The interviews lasted between 40–45 minutes. Equally, the sampled participants in the FGDs ought to be able to provide significant insights and points of view regarding the topic under study. The local council chairperson and community health workers acted as the "community gatekeepers" to the study site, thus facilitating and helping in identification and the recruitment of participants.

The FGDs were held as; two only male, one only female and one combined group for credibility purposes and exploration of gender dynamics that influence disease management (Table 1). The combined FGD comprised of twelve men and women in a ratio of 1:1. The FGDs included farm workers, farm mangers, livestock farmers and those that had recovered from brucellosis with one FGD from each subcounty; the combined FGD also had participants from each subcounty for purposes of getting a variation of insights on brucellosis from different people. This enhances validity and credibility of the findings. Conversations were held by two trained research assistants where one researcher facilitated the discussion while the other took notes. Each participant was assigned a numerical identifier, ranging from one to eight, which they would raise to signal their intention to contribute or respond during the discussion. The moderator called on participants using their assigned numbers, and respondents would restate their number before speaking. This approach ensured that all participants had the opportunity to share their thoughts, maintained confidentiality by avoiding the use of names, and allowed participant numbers to be referenced accurately in the transcript derived from the audio recording. The discussions were balanced in order to ensure that both men and women equally participated in the discussions and lasted approximately 1 hour. All data was recorded on an audio recorder with written consent from the participants and were later transcribed into notes.

Table 1. Participants in the KIIs and FGDs and their average ages.

| Participants | Female | Male | Average Age |
|---|---|---|---|
| Key Informants (n=15) | 5 | 10 | 37 |
| FGD1 (n=12) | | 12 | 36 |
| FGD2 (n=10) | 10 | | 45 |
| FGD3 (n=12) | 6 | 6 | 33 |
| FGD4 (n=8) | | 8 | 36 |

## Data analysis

The general principles and procedures of qualitative data analysis by Clark and Braun [30] were followed. The analysis involved systematically examining, familiarization, organizing, and coding segments of the interviews. The audio recordings were transcribed and the transcripts were read by CK and LNN in order to gain a general comprehension of the data. The codes were adjusted, supplemented, or eliminated based on their importance to the evolving data. Subsequently, these codes were categorized into preliminary themes aligned with the study's objectives. The transcripts were exported to Nvivo12 pro (QSR International). PA, who is the project supervisor, made the final decision on the codes, sub themes and themes (S2 File). The themes were then reviewed in relation to the codes and underlying data, and were further refined and explicitly defined [31]. The analysis was done using inductive and deductive analysis following the constructs of the SEHBM. Analysis of verbatim statements from participants. Key points and data brought up during interviews were also written down and triangulated with focus group discussion data. These were checked for accuracy and expanded for complete narratives as a way of complimenting and elaborating on the findings [32]. The qualitative findings were reported following the guidelines of the Consolidated Criteria for Reporting Qualitative Research (COREQ), (S3 File) This 32-item checklist facilitated the comprehensive reporting of key elements, including details about the research team, study methods, study context, findings, data analysis, and interpretations [33].

## Ethical consideration

The Uganda National Council of Science and Technology (Ref: SS2223ES) and the Makerere Social Sciences Research Ethics Committee (Ref: MAKSSREC 0823698) both reviewed and granted ethical approval for the study. Before the study started, the Chief Administrative Office (CAO) of the district gave their administrative clearance. Prior to performing key informant interviews and focus group discussions, each participant provided written informed consent. After transcription and analysis, identifiers were removed to protect the privacy and confidentiality of the information sources.

## Results

The findings presented below utilize the (SEMHB) constructs, together with the themes identified from the transcripts as illustrated (Table 2).

Table 2. Shows summary of themes and response that emerged out of the analysis.

| Level of influence | Emerging themes | Response |
|---|---|---|
| **Individual level** | Social Demographic Characteristic-Ethnicity | Ethnical attachment to cattle |
| | Source of income | Preference of milk due to being livestock herders |
| | Knowledge | The local construction of brucellosis and its etiology and spread |
| | Religious practices | Role of religious entities in brucellosis management |
| **Interpersonal** | Gendered Role distribution | Social constructed roles of both men and women |
| **Community** | Cultural norms and milking practices | Milking norms<br>Cultural notion of smearing animal products in preparation of a bride |
| | Cultural connotations and assisted births | Cultural significance of assisted births of cattle |
| | Family and power dynamic | Knowledge sharing and accessibility<br>Sharing of cattle for power and prestige |
| **Organizational** | Structural factors | Poor enforcement of hygiene practices<br>Proactive engagement of medical personnel with the community<br>Neglect of human brucellosis |

## Individual factors

These characteristics include elements such as the demographic characteristics like age, social economic status, knowledge and attitudes, skills, and beliefs. This holistic understanding of an individual's makeup helps us comprehend the nature of their behavior, acknowledging that various internal factors collectively contribute to shaping their actions and choices.

**Social demographic characteristics.** In the Nakasongola cattle corridor, brucellosis transmission affects both genders, with certain patterns associated with each as indicated in the interviews. As per key informant interviews, women are commonly more susceptible to brucellosis, although men are not immune. However, men's health-seeking behaviours, which typically involve seeking medical attention only when their condition becomes severe, might have implications for disease transmission and control. A male local leader noted,

*"Normally, I have seen more women than men maybe because men usually move with their illnesses, act strong and secretive and slow to report their ailments than the women"* (KII WB, P2-Local leader).

This excerpt is further emphasized by a medical officer who noted that,

*"Men usually have a poor health seeking behaviour, and personally if I see a man seated in the line claiming he is sick, I will attend to him very fast so by the time they come to hospital they are critically ill".* (KII WB.P4- medical officer)

The findings of the study also highlight Nakasongola cattle corridor as characterized by livestock farming, with both large and small-scale farmers playing a significant role. The key players in this agricultural landscape are those involved in the production and sale of milk. It was also noted that there is a strong relationship between the Nakasongola herders commonly known as the "*balalo*" and their cattle. This ethnical attachment to cattle emerged as another significant social factor linked to the transmission of brucellosis. Varied perspectives and the distinct cultural significance attributed to milk and cattle played a crucial role in influencing the transmission of brucellosis within specific ethnic groups. During the FGD with farmers, participants noted that convincing individuals from the "*balalo*" ethnic group to slaughter and bury their cattle in the event of a brucellosis infection was a challenging task. Similarly, requesting them to refrain from pouring or consuming the milk of an infected cow posed difficulties. The social connection to cattle surpasses that of family members and the broader community. A notable phenomenon in the district is the strong attachment that livestock farmers have to their cattle. For instance, in situations where both cattle and a family member are unwell, the farmers tend to prioritize the well-being of the livestock over that of the human family member as it was denoted in the FGDs.

*"Us the balalo, I can't say livestock farmers, the passion of the cows between the livestock farmers and the balalo is different, like if you slept with your wife and she complained of sickness and you found a cow sick, the attention you would give to the cow is different compared to the wife, you could even wake up at 3am and attend to a cow if its sick and yet tell your wife, to get a mere painkiller. A cow is not just an animal, the bondage between the balalo and their cows is very strong".* (FGD 4, P7).

The prioritization of cattle in this community might reflect their deep social significance. It is characterized as unique, and this suggests the potential for a strong connection between cattle and the community's social fabric.

**"Milk" as a way of life.** The study revealed that milk serves as a staple and is commonly consumed as breakfast, a beverage, and a food source within the livestock communities. Notably, it was observed that children, especially the younger ones, are often fed with raw milk due to cultural beliefs held by their parents. Consumption of raw milk, while a source of essential nutrients like protein, vitamins, and calories for children's development, might pose a potential risk of brucellosis transmission. In a key informant interview, a farm owner acknowledged the responsibility of ensuring that only

safe milk is either sold or retained for family consumption. Despite this responsibility, the owner highlighted the challenge of mothers not consistently boiling the milk before giving it to their children. There is a tendency of mothers sending their children to accompany their fathers to the farms during milking, and if a man refuses to provide milk, he may be perceived as prioritizing money over family welfare.

The consumption of raw milk developed as a significant dietary component for herders, particularly among the young males tasked with cattle herding responsibilities. Across all Focus Group Discussions (FGDs) and key informant interviews, it was consistently highlighted that these young males often lack the time to return home for meals or engage in cooking while working in the fields. Consequently, they rely on cow milk as a substantial source of sustenance. This observation was further validated during fieldwork, where herders explicitly mentioned that milk serves as a remedy for hunger, providing a feeling of fullness that lasts for an extended period. Additionally, the herders emphasized that the warmth of the milk contributes to maintaining body temperature, proving particularly beneficial during rainy periods and cold days.

*"I was born from a cattle farming family; I have lived and we have survived on cattle. Even if we are told that milk has brucellosis, milk is part of us. It does not matter whether you are educated on or not, drinking of raw milk is our identity. This is indoctrinated among us from birth. Milk keeps us warm, healthy and not go hungry. As you can see, how far is it from here to the house? How do you expect us to go back to the house for food yet there is food here, hahahaha"?* (FGD 1, P2).

**Knowledge on brucellosis transmission.** The local construction of brucellosis commonly known as "*omusujja gw'nte*" literally meaning" the disease of cows", presentation is overtly scary. One of the survivors noted that the pains especially in the back never ceased, the disease also manifested with chills in the evening and severe backache causing psychological trauma. The majority of the survivors indicated no knowledge on how they got the disease but were informed that brucellosis is got through drinking of milk and eating meat. However, some members of the community were questioning the cause of brucellosis being from milk and meat because they took all the measures but they still got the disease....

*"I was told brucellosis is got through drinking milk and eating meat but I have an in-law who does not eat meat or even drink milk but she also got brucellosis, so I wondered what I was being told. Also, I boil my milk thoroughly, I cook my meat until it is ready, so I got perturbed. I was also told to avoid being around cattle because it's also a route cause but how do you stay away from your cattle when there is no one to feed them or even graze them. The medical personnel were not very clear. I was told to stop drinking milk and eating meat but basing on the fact that my in-law got the disease yet she does not use any animal products, it led me to continue taking my milk and meat and also taking care of my cattle".* (FGD 2, P3)

The participants also showed concern that, though the disease is known as *omusujja gw'ente* locally, it is also found in other animals like goats and sheep. Meat from small ruminants like goats, sheep and pigs is the major source of meat in public spaces. A farmer in one of the FGDs noted that goats were found positive for brucellosis yet roasted goats' meat is a common delicacy especially at the social gatherings like weddings and bars.

**Religious and healthcare practices.** During data collection it was noted the religious affiliation of individuals, in this case Muslims, seems to have a noteworthy influence on their occupational risks, leading to increased exposure to brucellosis-infected animals. A key informant and a social leader noted Muslims are typically the ones responsible for slaughtering animals, excluding pigs. This group also includes a significant number of livestock traders, butchers, and other professions that require close interaction with cattle meant for slaughter. He said,

*"Such close and frequent contact with animals can increase the chances of exposure, leading to a higher risk of infection".* (KII, NB-P5- Social leader)

From the above insight, we see that the religious practices of Muslims, which often dictate their occupations, can inadvertently increase their exposure to animals infected with brucellosis, which could result in continuing transmission within the region.

### Interpersonal factors in the transmission of brucellosis

**Role distribution.** The women involvement and activities done within the cattle corridors poses a risky attribute to brucellosis. A key informant mentioned, women for instance in the abattoirs are involved in slaughter and collection of cattle body parts like the skin, skin, blood, and hooves. These animal body parts are specifically for sale in the local food markets. However, it was observed that the collection of these items is done without protective gear thus predisposing them to infection as observed during data collection and denoted by female veterinary extension worker.

*"Here at the abattoirs, the women are involved and collect skin, the hooves and the head of the cattle for sell in the food markets/ street foods. Economic and financial factors play a big role in this aspect as some women collect these items to improve their financial status. However, the blood is usually fed to the dogs, poultry and pigs at home". (*KII, NK 1-female veterinary extension worker)

Further still, women participation in familial activities and traditional processing of milk and milk products like making of ghee and the traditional ghee sauce commonly known as "*eshabwe*" predisposes them to brucellosis. Eshabwe is one of traditional delicacies in some ethnic groups and its consumption is significant. Likewise, the consumption of skimmed milk after the ghee is extracted commonly known as "*amacunda*" is believed to increase sexual arousal and libido among women.

The findings of the study also highlight a disproportion in livestock herders' access to training and their initiative in seeking information about brucellosis. An interesting finding emerged; that, while women in this community appear to utilize health facilities more often, men seem to be more likely to participate in disease management training programs. as it was denoted by a key informant;

*"Often times you see few women during the trainings maybe because of their familial obligations, they tend to be less informed on the prevention and spread of brucellosis because they don't attend trainings which creates a gap of awareness in the community about disease management, particularly brucellosis".* (KII, NB 3- male medical personnel).

The study suggests a gender gap in brucellosis knowledge, with women scoring lower than men on related assessments; yet it plays a pivotal role in both the transmission and prevention of diseases. Although the key informants like medical personnel mentioned and agreed that the brucellosis at the human side is neglected, they mentioned that rather the veterinary sector works a lot in terms of vaccination and sensitization. They mentioned that they have health talks at the hospitals but are reluctant to go to the village. The interventions and measures for control of transmission require funding and support yet the government and local government have not put in a lot of effort.

### Community factors in the transmission of brucellosis

**Cultural norms and gender disparities in the milking practices.** It was also observed and noted that there are some cultural norms at the production stage and on the farm. Principally, women were restricted from participating in milking, particularly during menstruation, placing men in the forefront of these activities. Men were identified as being affected mainly because of their constant and close proximity with the animals especially during milking, treatment and feeding. The involvement of men in such activities also rendered them vulnerable and susceptible to brucellosis. For instance, unclean milking methods (such inadequate hand washing and dirt in production areas) might also aid in the

spread of zoonotic illnesses. Furthermore, in the production stage or on the farm, women were primarily responsible for collecting milk. However, it was observed that the hygiene practices during milk collection and the storage conditions for the milk were suboptimal.

Although it was noted that men are majorly involved in livestock rearing, women were also observed and it was noted that they are involved in backyard livestock farming especially with small ruminants like goats and sheep. It was also noted that families with one or two cattle in the backyard, women were more involved in taking care of the cattle.

Furthermore, the cultural and social notion of bathing and smearing of milk products among the balalo women was also noted as a gendered social factor for transmission of brucellosis. For example, women in the FGD highlighted that prospective bride are traditionally secluded indoors for a duration of three or more months leading up to their weddings, a ritual known as "*ogutegura omugyenyi*". Throughout this ritual, they are regularly provided with gallons of milk to enhance their physique and beauty, and their skin is coated with ghee to achieve a smooth and softened texture. The drinking of raw milk continues even to the husbands' home in order to maintain her beauty. While these practices high-light the cultural importance attributed to milk and its derivatives, they also have the potential to influence the transmission of brucellosis.

*"To say that milk and ghee is bad, is like an insult. When you live here and are a livestock farmer, a mulalo, you do not talk about milk and ghee or any other thing related to cattle as risk because you don't know the importance of milk. The beauty of a mulalo woman is seen by the weight and soft skin. Once the woman comes to your home as a wife, it's the role of the man to look after the woman and maintain her beauty. A big, smooth skinned women indicate a well-cared for women. This increases our ego and show our masculinity roles"* (FGD 3, P2-).

**Cultural connotations and assisted births.** The study revealed that transmission and vulnerability of brucellosis among men constituted assisting in births of young calves, often without protective gears and had a cultural implication. A key informant mentioned that some ethnic groups associate wearing gloves during parturition with less attachment to cattle and contempt for the calf. This also included cleaning of cow dung and any other animal product.

A participant noted that,

*"I visited a family of the balalo and during my stay within the family, one day I was asked to clean dung from the compound of the home, I got a spade to shove off the dug, the elder of the family lamented, "ogu no mwiru manya" implying that I do not belong to the cattle keeping ethnical group commonly known as balalo".* (FGD 1, P5-).

**Family and power dynamics.** It was also observed that the patriarchal forces within families also facilitated the transmission of diseases in terms of knowledge sharing and access. In some families, it was observed that talking to a woman especially during interviews was after seeking permission from the man. Although these were interviews for research, it showcased a possibility of being the same even in sharing information and accessing information regarding health issues. Actually, women in the focus group noted that it's the men who usually go to the sub-counties in case of any meeting or sensitization program, the women and wives were usually not allowed there. During an interview, it was noted that this situation is intensified by the prevalent illiteracy among women, leading to a lack of awareness regarding their rights.

*"Women's access to knowledge is very important in the control of brucellosis in the livestock farming communities. Their involvement in decisions affecting their lives is limited, and their confidence in asserting their rights is particularly low mostly because they are not aware of their rights which intensifies their marginalization. The absence of influence among women in the cattle corridor communities has cast them primarily as victims of health concerns".* (KII WB 2-medical personnel).

Thus, gender inequality is believed to be firmly ingrained in these societies, as women face constraints preventing them from contributing to crucial discussions concerning community development, economic progress and food resources.

The findings also showcased that within the community, authority, impact, and the concept of "manhood" are all associated with possessing cattle. Social and cultural practices like sharing and exchange of cows during ceremonies like marriages were identified as points of showing off one's social status and interconnectedness. In this cultural practice people give out cows as gifts; this reciprocal exchange is known as "*empaano*," involving the rotational exchange of cows among friends, relatives, and in-laws. During feasts especially marriages; the farmers would frequently give out cattle and also butcher animals at home during these customary rituals without the use of meat inspection agencies or personnel.

*"While celebrating feasts, particularly during marriages, farmers often slaughter animals at home as part of these traditional rituals, without the involvement of meat inspection agencies or personnel and the more cattle slaughtered the more powerful one is."* (FGD 3, P6)

### Organizational factors for the transmission of brucellosis

**Structural aspects.** The study also revealed that another dimension influencing the transmission of brucellosis revolves around structural and organizational factors. Observations of poor hygiene practices, including collecting animal blood with bare hands and inadequate disposal of abortus especially at the slaughter houses was noted. This often has direct implications for the transmission of Brucella bacteria from infected animals to humans. These practices increase the likelihood of individuals coming into direct contact with the pathogen, leading to potential infections. A female veterinary extension worker noted that there was lack of strict measures of adherence to hygiene protocols, including the mandatory use of personal protective equipment (PPE), especially gloves, during the handling of animals and their bodily fluids.

Further, it was noted that there is a significant oversight in addressing human involvement, despite the fact that brucellosis impacts humans as it was highlighted by men in the focus group discussion.

*"Humans are mostly neglected yet the disease also affects humans but is not given the utmost attention. Also, interventions and measures require funding and support, we need to harmonize systems, for instance the one-health approach has failed to kick off yet it's very important though under human health, we usually have health talks at the hospitals but are still reluctant or slow to go to the community".* (KII, NK 4- Medical personnel)

## Discussion

The significance of gender and the social construction in disease transmission, particularly concerning zoonotic diseases is crucial in providing a vital perspective in the health discourse. This viewpoint forms the basis of our current study, which aimed to explore the interplay between gender dynamics, vulnerability and social construction in the management of brucellosis using the SEMHB. The study covered individual, interpersonal, community and organizational factors underpinning transmission of brucellosis identified in previous research. At the individual level, knowledge, attitudes, and practices regarding brucellosis play a critical role in determining susceptibility and response [34]. The complexity of brucellosis diagnosis and control is compounded by the diverse socio-cultural and environmental conditions across the region. Traditional cultural and gender roles dictate the activities men and women engage in, affecting their vulnerability to infectious disease [35]. Positioning these factors within wider social and ecological contexts allows for an analysis of how interactions between individuals, communities, and the environment influence disease transmission and gender-based vulnerabilities. Thus, our aim was to contribute to the formulation of more comprehensive and community-centered strategies to mitigate brucellosis transmission

The results in this study showed that farmers' practices such as the consumption of raw milk and undercooked meat increased their risk of brucellosis. This concurs with a study by Musallam et al. [36] that indicated a high risk of infection if

unpasteurized milk is consumed. The farmers practices could be because of accessibility, perceived nutritional value, taste preferences, and a variety of sociocultural factors, such as preferential food allocation within households as has been in literature [37]. The tradition of butchering and sharing of cattle at home during traditional ceremonies and customary rituals without the use of meat inspection agencies or personnel, further exacerbates this risk. This narrative unveils the manner in which communities engaged in livestock farming exist and pursue their way of life. Whilst these practices are frequently carried out at traditional ceremonies like marriages, they continue to put communities at higher risk of exposure because it has been linked to zoonotic disease prevalence in Africa [2]. Emphasis is put on the need for addressing cultural habits and focusing on public awareness about the health risks of consuming raw animal products. The finding of the study aligns with findings according to other reports [2], which denotes that zoonotic disease prevalence is linked to this widespread practice in African nations.

Our findings also showed that feeding of young children on raw fresh milk and colostrum from animals to boost their immunity is a practice that needs attention because it puts them at higher risk of acquiring brucellosis. As noted by Kansime and colleagues in 2014, the practice is common among pastoralist communities who frequently consume raw fresh milk [38]. It is imperative to inform communities, parents, and caregivers about the dangers of consuming raw milk and colostrum, while advocating for pasteurization as an effective method to reduce the risk of brucellosis and other foodborne infections.

The study identified a significant gap in knowledge about brucellosis, its transmission, causes and prevention, similar to results reported from other studies among agro-pastoralists in Uganda's cattle corridor [39]. If we are to attain optimal health as Abramowitz et al. [40] state, understanding how communities are aware of the condition is crucial to reducing its occurrence. Particularly, gendered disparities in access to information exacerbate vulnerability as women, who are frequently marginalized in decision-making processes, may have limited knowledge about brucellosis prevention and control measures. The lack of awareness contributes to the perpetuation of practices that facilitate transmission and is associated with ineffective precautionary behaviour needed to reduce exposure. The finding aligns with existing literature that emphasizes the importance of gender roles in disease transmission, particularly zoonotic diseases [41]. In the efforts to prevent and control brucellosis, there is a notable neglect of human involvement, despite the disease affecting humans. There is a gap in communication and education regarding brucellosis prevention and control, hindering the overall effectiveness of public health measures therefore, addressing these structural and organizational shortcomings could contribute to more comprehensive and community-focused strategies in combating the transmission of brucellosis.

At the interpersonal level, the study highlights a gender-based division of labor in farming activities, where women are restricted from participating in milking, especially during menstruation. The space context here contributes to a productive environment that manifests in activities shaping and reshaping the dynamics of brucellosis transmission. For instance, as the men are tasked with milking and treatment of the animals, women and children are responsible for cleaning the stalls as it was also noted according to one survey where 67% and 75% of women in the Walmera and Kersa Woredas, respectively, are in charge of stall cleaning. This indicates the significant role that women play in livestock farming [42] thus potentially influencing their exposure to brucellosis. Women's involvement in tasks such as cleaning livestock stalls and processing milk products, juxtaposed with men's engagement in off-farm production and milking processes, may increase their susceptibility to zoonotic diseases. A study in Zambia [43] indicated similar results such as men being involved in processing in factories and selling products for profit, while women are generally in charge of taking care of on-farm duties like gathering milk, traditional processing, and selling these products at the market.

The involvement of men in assisting of animal births without protective gear, due to cultural beliefs and practices adds a complexity to brucellosis transmission. Handling of the unborn calf or any animal product without gloves is a customary practice symbolic of the intimate connection between humans and their animals. The wearing of gloves, therefore was/is construed as an act of condescending contempt of their livelihood and livestock among pastoral farmers. In such cases, livestock farmers maybe exposed to brucellosis if flocks or herds are infected. This is a possibility that was discussed by Musallam et al. [36] in a study conducted in Jordan where only 6% of respondents reportedly wore protective gear while helping with livestock deliveries. However, this cultural belief and practice may influence the adoption of protective measures.

The study points out the constant and close proximity of men to the animals during milking, treatment, and feeding activities. The unclean milking methods, including inadequate hand washing and the presence of dirt in production areas and poor disposal of abortus, contribute to the spread of zoonotic illnesses thus increasing their vulnerability to brucellosis. This is consistent with findings from a study conducted in Jordan, wherein less than 10% of participants buried or burned the abandoned materials, while approximately 55% of participants fed the materials to dogs [36]. As Kansiime et al. [39] reported, inappropriate disposal of aborted fetuses highlights an additional area of concern where improvements in hygiene protocols could contribute to reducing the risk of disease transmission during the production process

While qualitative research provides rich description, the methodological limitations include the complexity of design which requires extended time and effort [44]. Reflections need to be made when using a qualitative approach due to the active role of the researcher. To address this limitation, we developed a semi-structured approach with pre-determined preliminary questions to ensure each of the participants experienced a consistent interview design. Also, the tools were pretested to ensure that the participants fully understood the intended question. As the lead researcher is not a native of the area, we had to know that perspectives and experiences are viewed from an outsider's perspective and we had to accept that we lacked first-hand information at the social construction and comprehension of management and transmission of brucellosis among the agro-pastoralists. This helped to view and consider the different perspectives participants hold owing to their social cultural and structural beliefs. In addition, data was taken from one district of which three sub-counties were chosen, hence not generalizable but often times this, qualitative research focuses in on the experience of a few individuals in an effort to infer aspects of a phenomenon involving many, which we demonstrate through their voice, their perspective, their lived experiences and their understanding.

## Conclusion

The ability of communities to avoid or control infections, adopt new ideas, embrace behavior change initiatives, and engage in health-seeking behavior are all crucial components of managing infectious diseases and this is facilitated or hindered by the social construction of the disease to include gendered individual, community, and social factors. From this study, it is now evident that the transmission of brucellosis is socially constructed through gender roles, socio-cultural norms, and power structures, with living conditions and spaces at various societal levels playing a crucial role in shaping individuals' susceptibility to infection. Gender-based vulnerabilities, influenced by socio-cultural practices and familial responsibilities, significantly contribute to exposure and disease spillover at individual, interpersonal, and community levels. Furthermore, inadequate knowledge dissemination and resistance to preventive measures emerge as structural factors perpetuating the persistence of brucellosis. These elements, intertwined with family dynamics and power relations, highlight the need for interventions that not only address individual behaviors but also tackle broader socio-cultural and institutional barriers to effective disease control. While individuals are responsible for implementing and maintaining the lifestyle changes required to minimize risk and promote health, all the levels of influence play a big role in determining individual behavior. Therefore, policies that are developed in alignment with the community's social context and values are more likely to be accepted, implemented, and sustained by the affected population.

## Supporting information

**S1 File. Interview guide.**
(DOCX)

**S2 File. Codebook.**
(XLSX)

**S3 File. COREQ-Checklist.**
(DOCX)

## Acknowledgments

We would like to thank all the participants for their support during the study, as well as the Nakasongola district administration.

## Author contributions

**Conceptualization:** Christine Tricia Kulabako, Stella Neema, Peter Atekyereza.

**Data curation:** Christine Tricia Kulabako, Peter Atekyereza.

**Formal analysis:** Christine Tricia Kulabako, Lesley Rose Ninsiima, Lydia Namakula Nabawanuka, Peter Atekyereza.

**Investigation:** Christine Tricia Kulabako.

**Methodology:** Christine Tricia Kulabako.

**Project administration:** Christine Tricia Kulabako.

**Supervision:** Stella Neema, James Muleme, Peter Atekyereza.

**Visualization:** Christine Tricia Kulabako.

**Writing – original draft:** Christine Tricia Kulabako.

**Writing – review & editing:** Christine Tricia Kulabako, Stella Neema, Lesley Rose Ninsiima, Jörn Klein, Lydia Namakula Nabawanuka, James Muleme, Javier Sánchez Romano, Peter Atekyereza.

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
