## [Decision Letter · Decision Letter 0]

21 Oct 2024

Dear Dr. Kulabako,

We look forward to receiving your revised manuscript.

Kind regards,

Avanti Dey, PhD

Staff Editor

PLOS ONE

Journal Requirements:

Additional Editor Comments (if provided):

Reviewers' comments:

Reviewer's Responses to Questions

**Comments to the Author**

1. Is the manuscript technically sound, and do the data support the conclusions?

Reviewer #1: Yes

Reviewer #2: Yes

Reviewer #3: Yes

2. Has the statistical analysis been performed appropriately and rigorously?

Reviewer #1: Yes

Reviewer #2: Yes

Reviewer #3: N/A

3. Have the authors made all data underlying the findings in their manuscript fully available?

Reviewer #1: Yes

Reviewer #2: Yes

Reviewer #3: Yes

4. Is the manuscript presented in an intelligible fashion and written in standard English?

Reviewer #1: Yes

Reviewer #2: Yes

Reviewer #3: Yes

Reviewer #1: Based on the article, I would say that it is generally able to be published in Journal PLOS ONE, but with some minor revisions to enhance its clarity and readability

The article presents a qualitative study that explores the social and cultural factors influencing the transmission of brucellosis in a rural community in Uganda. The study's methodology is sound, and the authors collected rich data through focus group discussions, key informant interviews, and field observations. Also, the findings suggest that cultural beliefs and practices play a significant role in the transmission of brucellosis, particularly in relation to the handling of cattle and the consumption of raw milk. The study highlights the importance of gender roles in disease transmission, as women are more likely to be involved in tasks that increase their exposure to brucellosis. Furthermore, the article also identifies gaps in communication and education regarding brucellosis prevention and control, which may contribute to the neglect of human involvement in disease transmission. On the other hand, the study's findings are consistent with existing literature on zoonotic diseases and the importance of addressing cultural and social factors in disease transmission.

The article is a qualitative research study, and the authors seem to have conducted a thorough analysis of the data. However, the writing style is quite dense and could be improved for clarity and readability.

The use of technical terms, such as "zoonotic diseases" and "omusujja gw'ente," may not be familiar to non-experts in the field. It would be helpful to provide definitions or explanations for these terms.

The study's findings are presented in a descriptive manner, but it would be beneficial to provide more specific statistics or numbers to support the claims.

The authors mention methodological limitations, but they do not provide enough details about their approach. It would be helpful to include more information about the research design, data collection methods, and sampling procedures.

However, the study has some limitations, including its small sample size (only three sub-counties were selected) and its reliance on self-reported data.

the study provides valuable insights into the social and cultural factors influencing the transmission of brucellosis in a rural community in Uganda. The findings highlight the importance of addressing cultural beliefs and practices that contribute to disease transmission, as well as improving communication and education regarding brucellosis prevention and control.

Reviewer #2: The manuscript entitled as “Gender and Cultural Aspects of Brucellosis Transmission and Management in

Nakasongola Cattle Corridor in Uganda” has been submitted to PLOS ONE for the publication. The manuscript present good review information on Gender and Cultural Aspects of Brucellosis Transmission in Uganda. This manuscript is nicely written with good scientific data, proper literature and explanations. Authors have tried to cover all necessary information related to the topic of manuscript. I would like to recommend this manuscript with one comment.

The authors can use the following articles to compare with their findings and discuss more about brucellosis in other countries and can include new treatment methods in the discussion.

https://doi.org/10.3389/fchem.2022.890252

https://doi.org/10.1007/s12011-019-01798-0

This will give complete better insight of selected topic.

Reviewer #3: Gender and Cultural Aspects of Brucellosis Transmission and Management in Nakasongola Cattle Corridor in Uganda

Abstract:

Results. It is common knowledge that practices such as consumption of raw milk, assisting in births rooted in social norms aid in transmission of brucellosis. What the study found on these in the study area is not clear in the abstract.

Conclusion: The issues raised are quite critical in brucellosis transmission and control. However, how these apply in the study were not elucidated.

Affiliations 1 & 2 should be well separated.

Introduction: The authors were unable to elucidate clearly the importance of ‘gender’ in the ‘Social Ecological Model of Human Behaviour (SEMHB)’ theoretical framework in creating a … that resulted in undertaking the study. While ‘vulnerability and social construction’ may be clear to me, the gender dynamics is not. The authors may wish to add a few lines improve the understanding of everyone.

Lines 66 and 67: ‘Brucella melitensis’ should be correctly spelt. ‘Brucella’ in the three different species mentioned should start with upper cases.

Line 77: ‘…existing literature have…’ There were more than one literature named.

Lines 127 – 129: Please, recast the sentence.

Line 146: ‘40-45 minutes’ should be written with spaces ’40 - 45 minutes’

Line 269: Should read ‘Milk keeps us warm, healthy and not…’ Although quoting a source, the whole excerpt was rephrased and therefore should be consistent. Same as line 282 : ‘until its’ should be ‘until it is’

334: ‘.. a lot effort ..’ should read ‘.. a lot of effort ..’

361: ‘..a mulalo women..’ ‘..a mulalo woman..’

Discussion

429 ; ‘This conquers’ should be written ‘This concurs’

443: Kansime et al, 2014 (32) should be written in conformity with other citations in the manuscript: Kansime et al. (32)

497: ‘,,understand the intended question’ to read ‘...understood the intended question’

Conclusion: The conclusion should reflect the findings of the work. The statements in the conclusions sound general and did not reflect their connection to the work.

**Do you want your identity to be public for this peer review?** For information about this choice, including consent withdrawal, please see our Privacy Policy

Reviewer #1: **Yes: ** Dr SeyedMousa Motavallihaghi

Reviewer #2: No

Reviewer #3: **Yes: ** Akwoba Ogugua

---

## [Author Response · Author response to Decision Letter 0]

8 Nov 2024

No Comment Response

1 Abstract: How data was analyzed was not reflected in the methods

Thank you for the observation, data analysis has been included in the abstract in lines 34-35

2 Results: It is common knowledge that practices such as consumption of raw milk, assisting in births rooted in social norms aid in transmission of brucellosis. What the study found on these in the study area is not clear in the abstract Thank you for the insight, this has been explicitly indicated showing how its rooted in culture and gendered. This has been indicated in lines 39-43

3 Conclusion: The issues raised are quite critical in brucellosis transmission and control. However, how these apply in the study were not elucidated Thank you for the observation, the conclusion in the abstract has been revised showing a connection with the study. These have been shown in lines 52-54 & 56-63

4 Affiliations 1 & 2 should be well separated.

Thank you, this has been separated in lines 7 and 8

5 Introduction: The authors were unable to elucidate clearly the importance of ‘gender’ in the ‘Social Ecological Model of Human Behaviour (SEMHB)’ theoretical framework in creating a … that resulted in undertaking the study. While ‘vulnerability and social construction’ may be clear to me, the gender dynamics is not. The authors may wish to add a few lines improve the understanding of everyone.

Thank you for your observation and insight. We agree that gender dynamics are a critical aspect of understanding vulnerability and social construction within this model, as they influence access to resources, social roles, and risk exposure across different ecological levels. We have therefore added sentences in lines 111-114 indicating the gender dimension of the SEMHB model

6 Lines 66 and 67: ‘Brucella melitensis’ should be correctly spelt. ‘Brucella’ in the three different species mentioned should start with upper cases.

Thank you for the observation, the spelling error and upper case have been edited in lines 77-78

7 Line 77: ‘…existing literature have…’ There were more than one literature named. This has been edited in line 89

8 Lines 127 – 129: Please, recast the sentence.

Thank you for your insight, the sentence has been rephrased in lines 139-141

9 Line 146: ‘40-45 minutes’ should be written with spaces ’40 - 45 minutes’

This has been worked on indicating the spaces in line 163

10 Line 269: Should read ‘Milk keeps us warm, healthy and not…’ Although quoting a source, the whole excerpt was rephrased and therefore should be consistent. Same as line 282: ‘until its’ should be ‘until it is’

Thank you, the excepts have been edited and corrected. The sentence on Milk keeps……. Has been revised in line 286

While

The sentence on I cook my meat until…… has been addressed in line 299

11 334: ‘.. a lot effort ..’ should read ‘.. a lot of effort ..’

This has been addressed in line 351

12 361: ‘...a mulalo women..’ ‘..a mulalo woman.. Thank you for the observation, it has been edited in line 378

13 429; ‘This conquers’ should be written ‘This concurs’

Thank you for the observation, it has been edited in line 446

14 443: Kansime et al, 2014 (32) should be written in conformity with other citations in the manuscript: Kansime et al. (32)

Thank you for the observation, it has been edited in line 460-461

15 497: ‘, understand the intended question’ to read ‘...understood the intended question’

This has been edited in line 515

16 Conclusion: The conclusion should reflect the findings of the work. The statements in the conclusions sound general and did not reflect their connection to the work.

Thank you for this insight, this has been revised highlighting the study findings thus a connection to the work in lines 525 - 551

---

## [Decision Letter · Decision Letter 1]

21 Jan 2025

Dear Dr. Kulabako,

Thank you for submitting your manuscript to PLOS ONE. After careful consideration, we feel that it has merit but does not fully meet PLOS ONE’s publication criteria as it currently stands. Therefore, we invite you to submit a revised version of the manuscript that addresses the points raised during the review process.

**Please ensure that you have clarified your methods as requested.**

https://journals.plos.org/plosone/s/submission-guidelines#loc-laboratory-protocols . Additionally, PLOS ONE offers an option for publishing peer-reviewed Lab Protocol articles, which describe protocols hosted on protocols.io. Read more information on sharing protocols at https://plos.org/protocols?utm_medium=editorial-email&utm_source=authorletters&utm_campaign=protocols .

We look forward to receiving your revised manuscript.

Kind regards,

Rebecca Lee Smith, D.V.M., M.S., Ph.D.

Academic Editor

PLOS ONE

**Journal Requirements:**

Reviewers' comments:

Reviewer's Responses to Questions

**Comments to the Author**

Reviewer #2: All comments have been addressed

Reviewer #3: All comments have been addressed

Reviewer #4: All comments have been addressed

2. Is the manuscript technically sound, and do the data support the conclusions?

Reviewer #2: Yes

Reviewer #3: Yes

Reviewer #4: Yes

3. Has the statistical analysis been performed appropriately and rigorously?

Reviewer #2: Yes

Reviewer #3: Yes

Reviewer #4: N/A

4. Have the authors made all data underlying the findings in their manuscript fully available?

Reviewer #2: Yes

Reviewer #3: Yes

Reviewer #4: No

5. Is the manuscript presented in an intelligible fashion and written in standard English?

Reviewer #2: (No Response)

Reviewer #3: Yes

Reviewer #4: Yes

**Reviewer #2: ** Dear authors

According to the amendments made, this manuscript is suitable for publication.

Best wishes

**Reviewer #3: ** The authors have effected the corrections pointed out. The manuscript has improved and can be published in the present state.

**Reviewer #4: ** To editor,

This manuscript explores a critical public health issue that affecting human health negatively. The study highlighted on the brucellosis transmission vulnerability with gender dynamics, and sociocultural construction that can inform targeted interventions for the development to control measures and management strategies that will contribute to improve health outcomes for affected population especially the pastoralist communities in Africa.

That said, I would like to offer the following comments and suggestions:

1. Abstract

In line 35 and 36, the author write about the use of deductive and inductive approaches is overlapping.

2. Introduction

Okay.

3. Methods

Methods section is well written.

- However, the author need to revise about the sample size part. The authors did not explain about the justification of 4 FGDs and 15 KIIs scientifically. In addition, 1 FGD conducted with combined group but why it was not equal or shy it was combined is not explained. As the study focused on the gender dynamics, vulnerability and social construction in the management of brucellosis, the justification of sampling size and sampling strategy would be more evident.

- Additionally, the author can revise it as the study population and sampling strategies. Because, how the study participants recruited, participants rights, interview guidelines were developed did not describe. Better to revise it into separate part like sample size and sampling strategies and then data collection technique and procedures. Additionally, it would be more evident if the authors follow the COREQ.

- In line191, the author present ‘the context and interpretation……..enhanced by observation ….and field notes made during the interview…. is not mentioned in the data collection as a technique. In line 171-172, the author present that key points were brought up….and triangulated with observation which should move into data analysis.

- The triangulation issues are not only confined with the comparing data with key points (line 171-172), it would be participants, methods like discussion data with interview and/or observation. Suggesting to follow a citation.

- Line 175-179 suggesting move to the end of discussion as this writing are seemed as limitations of study.

- Data analysis part is not articulated well. How the audio data transcribed, familiar with the data, code list developed and updated are not explained? The authors started directly with the thematic analysis and then talk about the Nvivo transformation and so on.

- Additionally, how the code list entail following both deductive and inductive way? Need to explain.

4. Results

Okay but I would ask to include the interview number in the quotes.

5. Discussion

- In line 435, the authors just mentioned about SEMHB model but did not elaborated any in the discussion section. Although the findings are mostly interlinked with the model but did not discussed.

- In line 505, the author write: To address this limitation, ‘I’ developed… This is might be typo. Because in the methods section, the author addresses the term ‘We’. Need to revise one addressing style.

**Do you want your identity to be public for this peer review?** For information about this choice, including consent withdrawal, please see our Privacy Policy

Reviewer #2: **Yes: ** Dr Seyed Mostafa Hosseini

Reviewer #3: **Yes: ** Akwoba Joseph Ogugua

Reviewer #4: **Yes: ** Dr. Md. Shahgahan Miah

---

## [Author Response · Author response to Decision Letter 1]

24 Jan 2025

No Comment Response

1 Abstract: In line 35 and 36, the author write about the use of deductive and inductive approaches is overlapping.

Thank you for your observation, this has been edited in line 36

2 Introduction: Okay.

Thank you for the comment

3 Methods

Methods section is well written.

- However, the author need to revise about the sample size part. The authors did not explain about the justification of 4 FGDs and 15 KIIs scientifically. In addition, 1 FGD conducted with combined group but why it was not equal or shy it was combined is not explained. As the study focused on the gender dynamics, vulnerability and social construction in the management of brucellosis, the justification of sampling size and sampling strategy would be more evident.

Thank you for the insight. Scientific justification of 4FGDs and 15 KIIs has been explained. The sample size was based on participant saturation using maximum variation strategy, line 143-145.

The reason for unequal FGDs has also been included, Line 155-157

4 Additionally, the author can revise it as the study population and sampling strategies. Because, how the study participants recruited, participants rights, interview guidelines were developed did not describe. Better to revise it into separate part like sample size and sampling strategies and then data collection technique and procedures. Additionally, it would be more evident if the authors follow the COREQ.

Thank you, this has been revised, recruitment and how interview guidelines were developed has been included, line 146-149.

168-170 and 175-180. Guidance of the COREQ has been included, Line 210-213.

5 In line191, the author present ‘the context and interpretation……..enhanced by observation ….and field notes made during the interview…. is not mentioned in the data collection as a technique. In line 171-172, the author present that key points were brought up….and triangulated with observation which should move into data analysis. Thank you for your observation and insight. Observation has been deleted and moved to data analysis, Line 203-206

6 The triangulation issues are not only confined with the comparing data with key points (line 171-172), it would be participants, methods like discussion data with interview and/or observation. Suggesting to follow a citation.

This has been revised to include a citation, Line 202

7 Line 175-179 suggesting move to the end of discussion as this writing are seemed as limitations of study.

Thank you for the insight, this has been moved the discussion, ine 520-524

8 Data analysis part is not articulated well. How the audio data transcribed, familiar with the data, code list developed and updated are not explained? The authors started directly with the thematic analysis and then talk about the Nvivo transformation and so on.

Thank you for the observation, this has been revised, Line 200-203

9 Additionally, how the code list entail following both deductive and inductive way? Need to explain.

Revised in Line 200-201

10 Results

Okay but I would ask to include the interview number in the quotes.

Thank you, interview numbers have been included on the quotes

11 Discussion

In line 435, the authors just mentioned about SEMHB model but did not elaborate any in the discussion section. Although the findings are mostly interlinked with the model but did not discuss.

Thank you for the insight, an elaboration of the SEMHB has been included, Line 457-465

12 In line 505, the author writes: To address this limitation, ‘I’ developed… This is might be typo. Because in the methods section, the author addresses the term ‘We’. Need to revise one addressing style.

This has been edited, Line 523

---

## [Decision Letter · Decision Letter 2]

18 Feb 2025

Gender and Cultural Aspects of Brucellosis Transmission and Management in Nakasongola Cattle Corridor in Uganda

PONE-D-24-16528R2

Dear Dr. Kulabako,

We’re pleased to inform you that your manuscript has been judged scientifically suitable for publication and will be formally accepted for publication once it meets all outstanding technical requirements.

Kind regards,

Rebecca Lee Smith, D.V.M., M.S., Ph.D.

Academic Editor

PLOS ONE

Additional Editor Comments (optional):

Reviewers' comments:

Reviewer's Responses to Questions

**Comments to the Author**

Reviewer #4: All comments have been addressed

2. Is the manuscript technically sound, and do the data support the conclusions?

Reviewer #4: Yes

3. Has the statistical analysis been performed appropriately and rigorously?

Reviewer #4: N/A

4. Have the authors made all data underlying the findings in their manuscript fully available?

Reviewer #4: Yes

5. Is the manuscript presented in an intelligible fashion and written in standard English?

Reviewer #4: Yes

Reviewer #4: Thank you so much for addressing all the comments well and recommending the manuscript to the editor for publication.

**Do you want your identity to be public for this peer review?** For information about this choice, including consent withdrawal, please see our Privacy Policy

Reviewer #4: **Yes: ** Dr. Md. Shahgahan Miah

---

## [Editor Report · Acceptance letter]

PONE-D-24-16528R2

PLOS ONE

Dear Dr. Kulabako,

I'm pleased to inform you that your manuscript has been deemed suitable for publication in PLOS ONE. Congratulations! Your manuscript is now being handed over to our production team.

Kind regards,

on behalf of

Dr. Rebecca Lee Smith

Academic Editor

PLOS ONE